# Analysis of Rotational Stiffness of the Timber Frame Connection

**Marek Johanides** [1],* , **Lenka Kubíncová** [1], **David Mikolášek** [1], **Antonín Lokaj** [1] , **Oldřich Sucharda** [2] and **Petr Mynarčík** [3]

1   Department of Structures, Faculty of Civil Engineering, VSB—Technical University of Ostrava,
    708 00 Ostrava-Poruba, Czech Republic; lenka.kubincova@vsb.cz (L.K.); david.mikolasek@vsb.cz (D.M.);
    antonin.lokaj@vsb.cz (A.L.)
2   Department of Building Materials and Diagnostics of Structures, Faculty of Civil Engineering,
    VSB—Technical University of Ostrava, 708 00 Ostrava-Poruba, Czech Republic; oldrich.sucharda@vsb.cz
3   Department Centre of Building Experiments, Faculty of Civil Engineering, VSB—Technical University of
    Ostrava, 708 00 Ostrava-Poruba, Czech Republic; petr.mynarcik@vsb.cz
*   Correspondence: marek.johanides@vsb.cz

**Abstract:** Initially, timber was considered only as an easily accessible and processable material in nature; however, its excellent properties have since become better understood. During the discovery of new building materials and thanks to new technological development processes, industrial processing technologies and gradually drastically decreasing forest areas, wood has become an increasingly neglected material. Load-bearing structures are made mostly of reinforced concrete or steel elements. However, ecological changes, the obvious problems associated with environmental pollution and climate change, are drawing increasing attention to the importance of environmental awareness. These factors are attracting increased attention to wood as a building material. The increased demand for timber as a building material offers the possibility of improving its mechanical and physical properties, and so new wood-based composite materials or new joints of timber structures are being developed to ensure a better load capacity and stiffness of the structure. Therefore, this article deals with the improvement of the frame connection of the timber frame column and a diaphragm beam using mechanical fasteners. In common practice, bolts or a combination of bolts and pins are used for this type of connection. The subject of the research and its motivation was to replace these commonly used fasteners with more modern ones to shorten and simplify the assembly time and to improve the load capacity and rigidity of this type of frame connection.

**Keywords:** rotational stiffness; frame connection; screws; glued laminated timber; numerical model; FEM

## 1. Introduction

Wood is a material with highly variable properties [1] and is the only renewable material that is born in nature and disappears without negative consequences for the environment; furthermore, it can be used in construction for the implementation of all load-bearing structures. Among the most influential factors that cause this dispersion of properties is the species of timber and the location in which the timber grew. The soil, climate, altitude, the time of the year in which the tree was cut and, last but not least, the subsequent method and quality of processing have a major impact on the characteristics of timber. In the case of wood as a structural material, either grown wood, which is obtained by cutting from a trunk of coniferous or broadleaved timber [2], or wood-based material such as glued laminated timber [3], which is made by gluing timber laminations to the required size and shape, are considered. The timber exhibits different physical and mechanical properties in mutually perpendicular directions. This means that the characteristics observed parallel to the fibers are different from the properties observed perpendicular to the fibers. Timber has the greatest strength and stiffness and the least deformation due to moisture and temperature in the direction parallel to fibers. The mechanical properties of timber reduce its ability to withstand external loads. In this

context, it is necessary to distinguish between the properties of trouble-free timber (wood mass) and construction timber. The properties of trouble-free wood show a relatively considerable dispersion, which is most amplified in the case of constructional timber due to the influence of growth inhomogeneities. Wood is an anisotropic material in terms of its mechanical properties, but for the purpose of calculations or numerical modeling, it can be considered to be a material that is rectangularly or cylindrically orthotropic [1].

One of the uses of timber is the construction of hall buildings, which are very popular due to their design and are very important in civil engineering. The main advantage of hall buildings is the roofing of a large span without disturbing the interior layout of the building, so these buildings are suitable, for example, for the manufacturing industry, recreational facilities or for agricultural buildings. Steel is often used as the material for the construction of hall buildings [4–6]; however, this is disadvantageous due to its high density and low fire resistance [7,8]. In addition to the mentioned factors in the selection of construction material for hall objects, a major emphasis is also placed on the use of natural and esthetic materials [9,10]. This is typical, for example, in the construction of sports facilities. For these reasons, timber structures are increasingly used for construction.

A designer usually does not have the data available for the experimental testing of their design. This is mainly due to the financial costs and time required for an experiment. For this reason, it is most common for a designer to create a numerical or analytical model based on the recommended valid standards and scientific literature. In practice, however, the selection of the incorrect connection often leads to the oversizing of constructional members and thus to an uneconomical design of the building.

The frame connection of these structural elements is most often used for the connection of frame columns and diaphragm beams [11–13]. This connection of elements is one of the most important areas of the design of timber structures and the issue of design and assessment of the connections of timber structures as it fundamentally affects the overall composition of the supporting structure and the dimensions of the main supporting elements [14,15]. Thanks to the optimization of this area, significant material savings can be achieved during construction, which will reduce the price and complexity of the construction.

The load capacity and stiffness of connections are quite often a decisive factor for the design and operation of structures, especially in construction with larger spans, where the connection is heavily stressed. Connections of timber frame elements can be solved in several ways using glued connections [16–18]; for example, using glued steel rods. Another option is to create a frame connection from a frame column which is arranged in a V-shape [19]. The most commonly used type of connection is a frame connection of the frame column and diaphragm beam using mechanical pin-type fasteners [20].

The subject of this article is an experiment aimed at the frame connection of the frame column and diaphragm beam by mechanical pin–type fasteners. The constructional material of the frame column and diaphragm beam is glued laminated timber, while the frame connection is created using high-strength Rothoblass fully-thread screws. Modern high-strength screws are currently also used as a semi-rigid coupling means for wood-concrete ceilings, where they are an alternative to fastening with a glued steel bar [21,22]. In this type of connection, fasteners bolts or a combination of bolts and pins are commonly used. The use of screws is not common in practice and it was, therefore, the motivation of this work to deal with this type of fasteners for more detailed analysis in order to determine how the structure as a whole behaves, the load capacity of the frame and its connection and rotational stiffness. This is important for the redistribution of internal forces in the structural system of the bar model.

The failure of this type of connection with a frame column and diaphragm beam using standard fasteners (i.e., bolts or combination of bolts and pins) occurs under tension perpendicular to the fibers in the upper corner of the diaphragm beam. At this point, the highest tensile stress occurs from fasteners and leads to the subsequent destruction of the

diaphragm beam [23]. For comparison, it was interesting to observe this location on the diaphragm beam and the total failure of the connection formed from fully threaded screws.

There is currently no standard procedure for determining the load capacity and rotational stiffness of this connection. Therefore, the analytical determination of these values was based on the literature and articles [24–26].

## 2. Materials and Methods

### 2.1. Description of Construction and Geometry

The structural system of the experiment consisted of the bending rigid connection of a frame column and diaphragm beam by metal mechanical pin-type fasteners. The frame column had a cross-section of 180/700 mm and a timber class of GL24h. The diaphragm beam cross-section was $2 \times 120/700$ mm with a timber class of GL24h. The components were connected by fully threaded screws Rothoblaas VGS11400; the external thread diameter was 11 mm and screw length was 400 mm. The layout of these screws was on two symmetrically concentrated circles, where circle 1 had a radius $r_1$ of 273 mm with 24 screws. Circle 2 had a radius $r_2$ of 218 mm with 20 screws. The detailed locations of frame connection fasteners are shown in Figure 1.

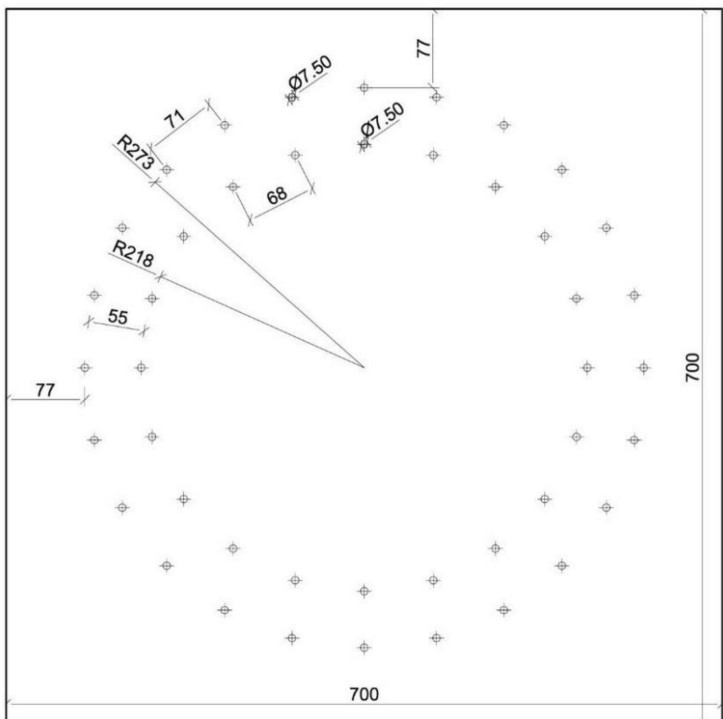

**Figure 1.** The detail of fasteners geometry at frame connection (Scale 1:7).

In order to carry out this experiment, it was necessary to create the boundary conditions of the structure. To ensure the correct boundary conditions, a steel structure was made (see Figures 2–4). The steel mounting was anchored to a reinforced 450 mm thick concrete slab.

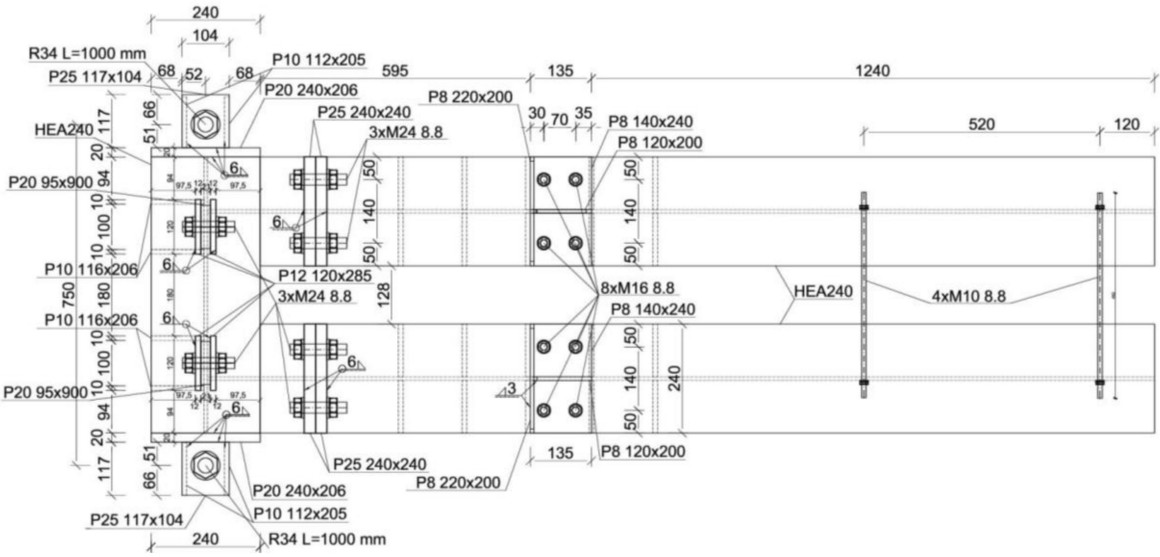

**Figure 2.** Floor plan of steel mounting (Scale 1:25).

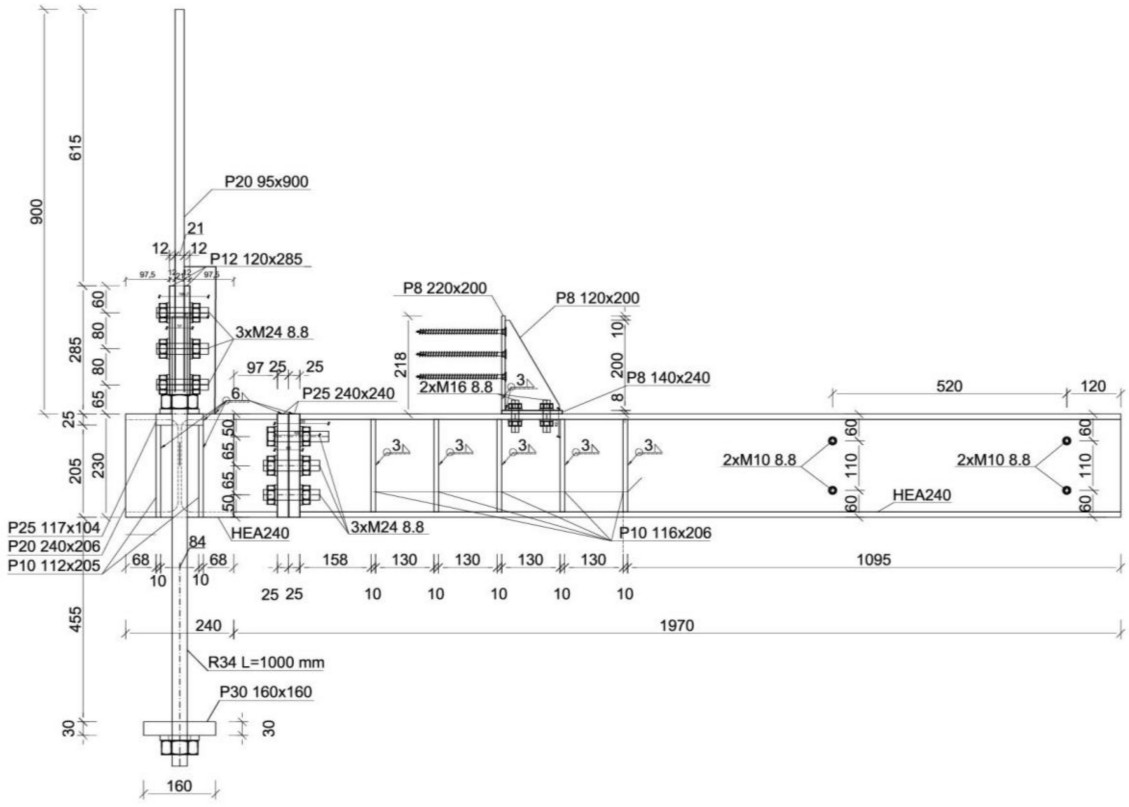

**Figure 3.** Side view of steel mounting (Scale 1:25).

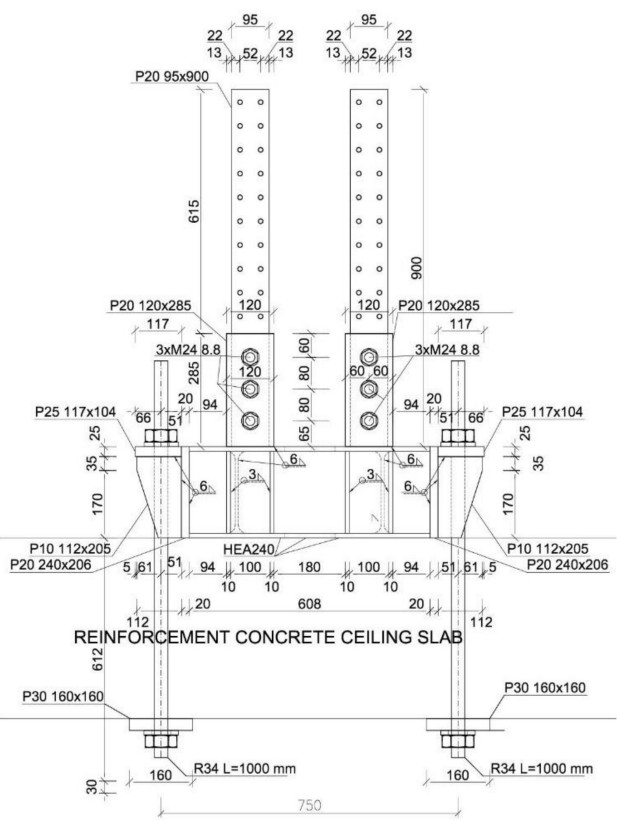

**Figure 4.** Back view of steel mounting (Scale 1:25).

The frame column of the frame structure was fitted into the mounting steel structure using sheets and screws, as shown in Figure 5.

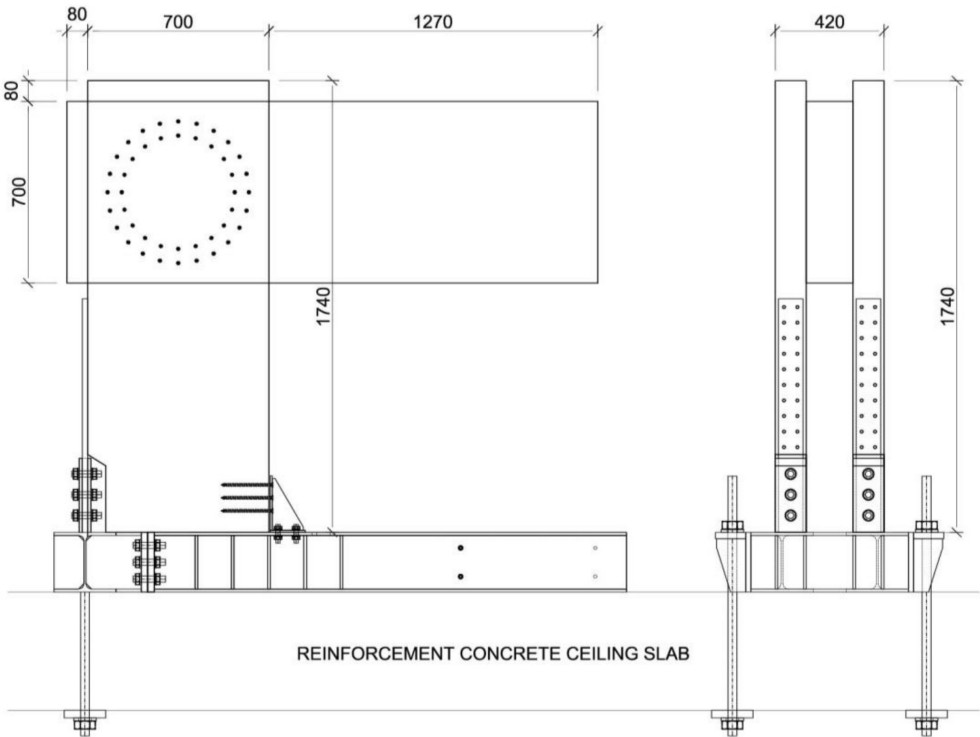

**Figure 5.** Design test system (Scale 1:25).

### 2.2. Experimental Measurement

The experiment was aimed at the determination of the load-bearing capacity and rotational stiffness of the frame connection of the frame column and diaphragm beam. In order to determine rotational stiffness, it is necessary to determine the deformation of individual parts of the structure. In order to monitor the deformation of the structure, two deformation sensors were mounted on the diaphragm beam according to the diagram shown in Figure 6. Specifically, a potentiometer sensor was used that is classified in the category of resistance sensors. A Type TR-0100 spring-loaded linear potentiometer position sensor, as shown in Figure 6, was used, which has a linear deviation of 0.01 mm with a maximum lifting length of 100 mm.

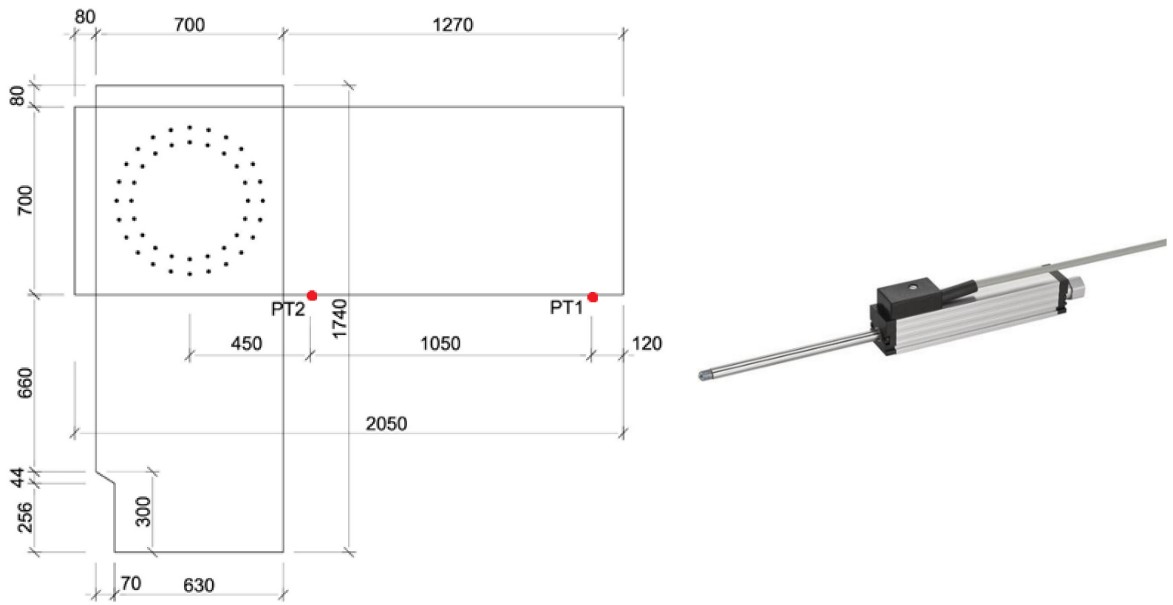

**Figure 6.** From left: Location of potentiometers (Scale 1:25), linear potentiometer [27].

#### 2.2.1. Description of the Test Machine

The experiment was performed in the Centre of Building Experiments, VSB (Technical University of Ostrava) on a hydraulic servo cylinder, which allows tensile and pressure static and dynamic tests. The maximum force that the electrohydraulic cylinders of the testing machine can exert is 400 kN, which was sufficient for testing the frame connection to failure.

#### 2.2.2. Description of the Frame Connection Loading

The loading was realized in several stages by deformation. The structure was first loaded to about 30% of its characteristic load capacity according to the standard [28], and then it was subsequently lightened. A second cycle followed in which the load was about 60% of the characteristic value of load capacity, followed by lightening. The last cycle was loading until failure of the connection. The load action is shown in Figure 7.

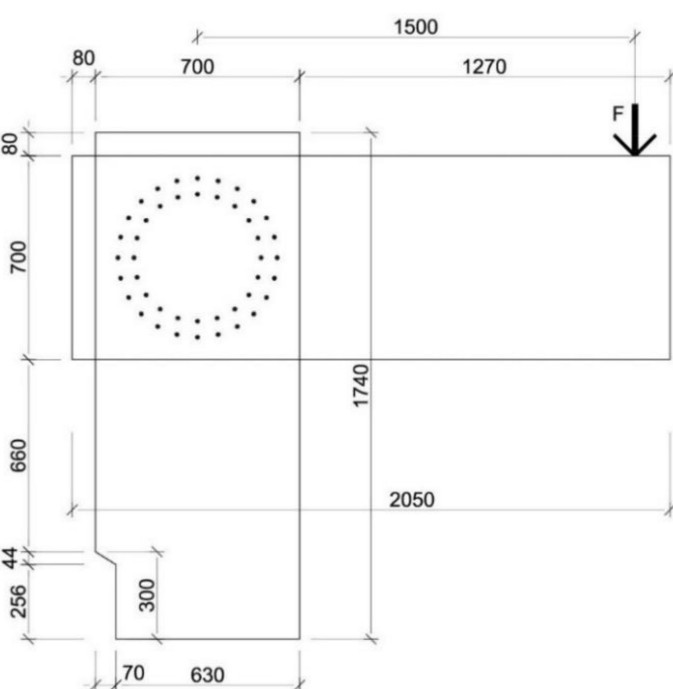

**Figure 7.** Force action scheme (Scale 1:25).

### 2.3. Analytical Calculation of Frame Connection Rotational Stiffness

The analytical method for calculating the rotational stiffness of the frame connection was implemented for the validation of analytical and experimental results.

To calculate the internal forces, it is important to correctly describe the rigidity of the connections. The main characteristic of the rigidity of the connection of timber structures is the slip modulus $K_{ser}$. This value expresses the displacement of the fastener from the given shear force in the shear surface and the axial force. The torsional stiffness $K_r$ expresses the rotation from the bending moment. The moduli of displacement $K_{ser}$ and $K_r$ were calculated according to [28] and the relevant literature [24,29,30].

#### 2.3.1. Calculation of Translational Stiffness

For a single shear laterally stressed screw, the bolt applies a force according to the equation

$$K_{ser} = \frac{\rho_m^{1.5} \cdot d}{23} \tag{1}$$

where

$\rho_m$    is the average value of density of the connected timber member;
$d$    is the fastener diameter.

Eurocode 5 allows the value $K_{ser}$ to be doubled for a steel-to-timber connection.
For axially loaded screws, we applied the formula

$$K_{ser} = 780 \cdot d^{0.2} \cdot l_{ef}^{0.7} \tag{2}$$

where

$l_e$    is the effective length of screw penetration;
$d$    is the screw diameter.

The design value of the instantaneous slip modulus is calculated by

$$K_u = \frac{2}{3} \cdot K_{ser} \tag{3}$$

The translational stiffness of the connection is given by the sum of the slip modules of the individual fasteners. For multiple shear connections with n fasteners in the ULS, the following relation applies:

$$K_{t,u} = 2 \cdot n \cdot K_u \tag{4}$$

where

$n$     is the number of fasteners;
$K_u$    is the translational stiffness of the fastener (3).

### 2.3.2. Calculation of Rotational Stiffness

The calculations of the torsional stiffness $K_r$ of the flexible connection can be determined using the slip module of fastener $K_u$ in the case of ULS, or the slip module $K_{ser}$ of the fasteners in the case of SLS by the formula

$$K_{r,u} = \sum_{i=1}^{n} K_u \cdot r_i^2 \operatorname{resp} K_{r,ser} = \sum_{i=1}^{n} K_{ser} \cdot r_i^2 \tag{5}$$

where $K_u$ and $K_{ser}$ are the slip modules for ULS and SLS (2) (3).

For the torsional stiffness of a double shear connection with $n$ identical fasteners in the ULS, we apply the following formula (it can also be analogously calculated in SLS):

$$K_{r,u} = 2 \cdot \left( K_u \cdot r_1^2 + K_u \cdot r_2^2 + K_u \cdot r_3^2 + \ldots K_u \cdot r_n^2 \right) \tag{6}$$

### *2.4. Numerical Model*

Before the experimental test, numerical models of the frame connection were developed in the software SCIA Engineer and ANSYS$^{TM}$, as shown in Figure 8. Input data fpr the numerical models were obtained by calculation according to the recommendations given in [28–30]. Figure 9 shows the deformed fasteners from the numerical model and the experimental testing. Figure 10 shows this deformed fastener in a detailed view.

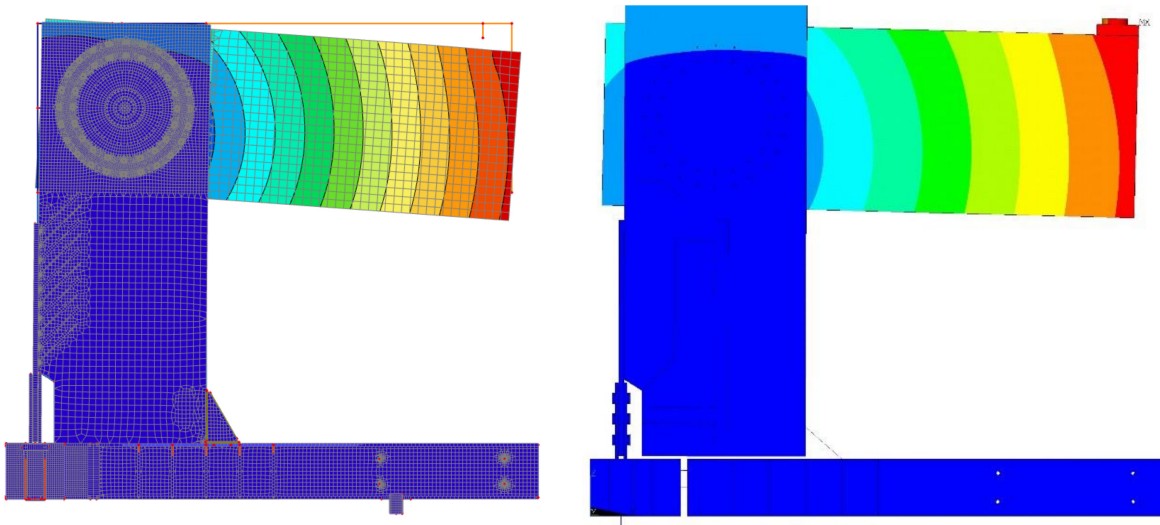

**Figure 8.** Deformation in the numerical model of SCIA Engineer (**left**), deformation in the numerical model of ANSYS$^{TM}$ (**right**).

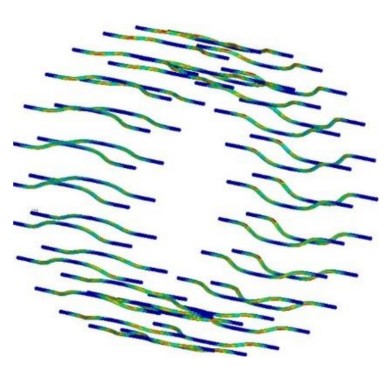 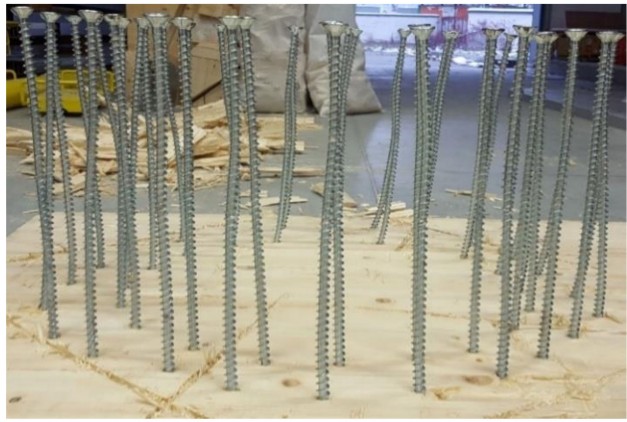

**Figure 9.** Deformed fasteners in the numerical model of ANSYS^{TM} (**left**), deformed fasteners after experimental testing (**right**).

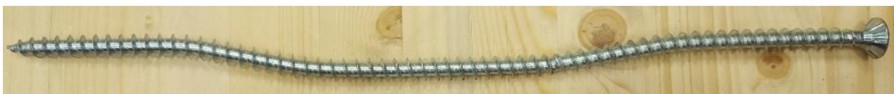

**Figure 10.** Deformed fastener after experimental testing.

### 3. Results

#### 3.1. Determination of Rotational Stiffness of the Connection

The determination of the manual calculation of the rotational stiffness of the connection was performed on the basis of Equations (1)–(6). Figure 11 shows the deformations of the potentiometers that were located on the experimental structure according to the scheme in Figure 6. Figure 12 shows the subtracted data of potentiometers PT2–PT1.

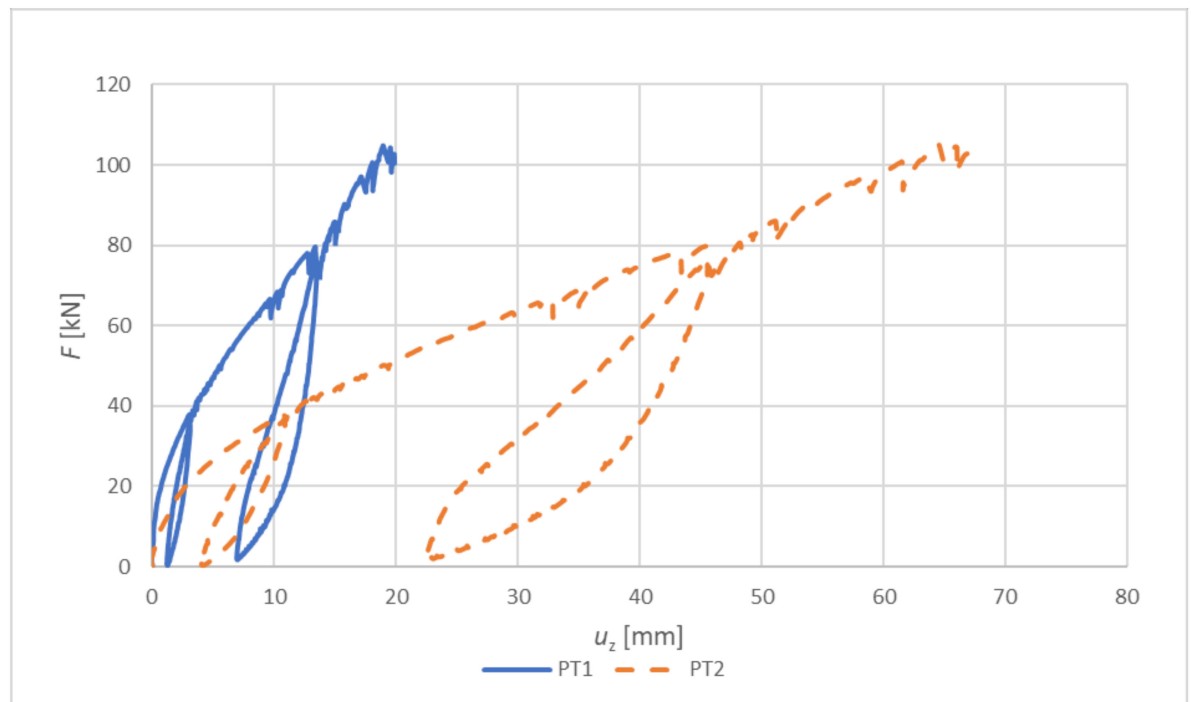

**Figure 11.** Deformation of potentiometers.

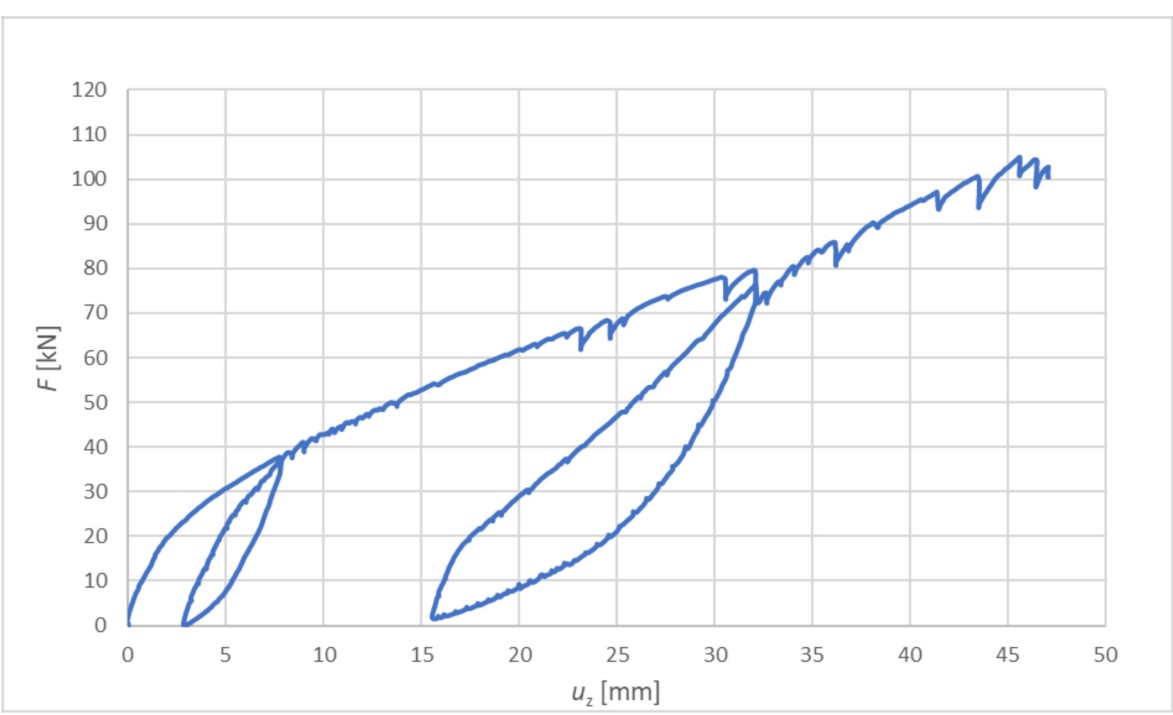

**Figure 12.** Subtracting values of potentiometers PT2–PT1.

Figure 13 shows the dependence of the rotational stiffness of the connection of the frame column and diaphragm beam on the force acting at the end of the diaphragm beam.

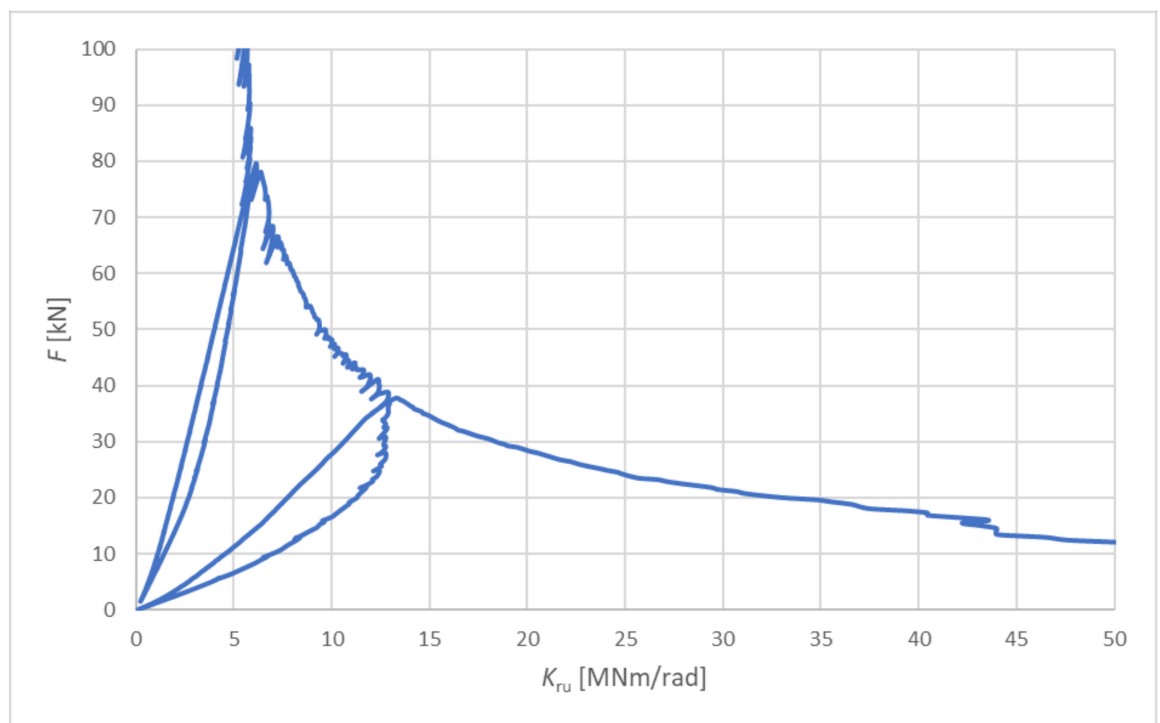

**Figure 13.** Dependence of rotational stiffness on the load without considering the decline of the frame column.

Due to the fact that this experimental test of the frame column and diaphragm beam was the first of a series of such tests, it was a so-called "zero test" (its task was to tune the boundary conditions and the method of measuring deformations and force); in this test, the potentiometer PT3, which measured the horizontal deflection of frame column under load,

was not present. Determining the rotational stiffness of the connection is impossible because we do not know the actual deflection of the frame column at a given load. Figure 14 describes the dependence of the rotational stiffness on the deformation caused by the inclination of the column. This graph shows the vertical deformation that would be caused by the horizontal deformation of the column on the vertical axis. The rotational stiffness, the horizontal axis of the graph, was then calculated for a specific load of $F = 31.96$ kN. This value corresponds to one-third of the design load determined based on [28], changing the total of the deformation. This consists of subtracting the potentiometers PT2–PT1, subtracting the deformation from the bending moment and subtracting the deformation from the shear force. Deformation from the bending moment of the frame column and deformation from the shear force is not deduced in this case, because it is included in the horizontal displacement of the frame column.

The dashed orange vertical line shows the characteristic value of rotational stiffness $k_r = 19.40$ MNm/rad calculated according to [24], and the dotted grey vertical curve shows the design value of rotational stiffness, which represents the value $k_{r,u} = 12.90$ MNm/rad.

Thus, in this graph, only one value is correct, which corresponds to the rotational stiffness for load force $F = 31.96$ kN and the actual deformation of the end of the diaphragm beam after subtracting all the above-mentioned deformations.

As a result of the measurement in this case, to achieve the actual rotational stiffness of the connection, which is close to the characteristic value according to [24] $k_r = 19.40$ MNm/rad, a real deflection of the frame column at a given load of about 2 mm is required. The value of 2 mm is realistic for this type of structural system and load effect, but to verify this, it is necessary to perform further tests to determine the value of deflection of the frame column.

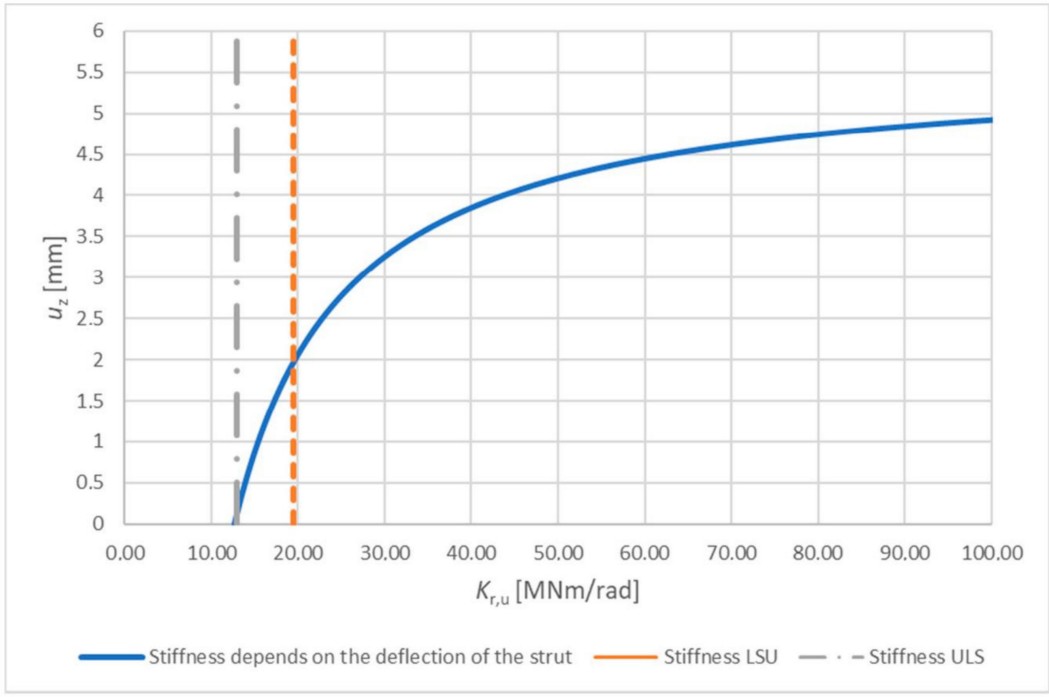

**Figure 14.** Dependence of stiffness on deformation caused by the decline of the frame column.

### 3.2. Determination of Load Capacity of the Connection

Figure 15 shows the dependence of the applied deformation load and the displacement of the extensometers (dotted and dot–dashed curve, displacement at the end of the diaphragm beam). The dotted curve was formed according to the analytical relationship of the force method using a rotational stiffness of $K_{r,ser} = 19.50$ MNm/rad (calculated as a characteristic value according to [28]). The dot–dashed curve is the result of a linear numeric model in ANSYS, and the double-dotted curve is the result of a nonlinear numeric

model created in ANSYS. By folding the linear line (solid line) over the load and unload cycle, it is possible to estimate the rigidity of the entire frame construction system, including the steel structure providing the boundary conditions. When comparing the slope of the experimental test line and the numerical model, we can observe a relatively good agreement in terms of rotational stiffness.

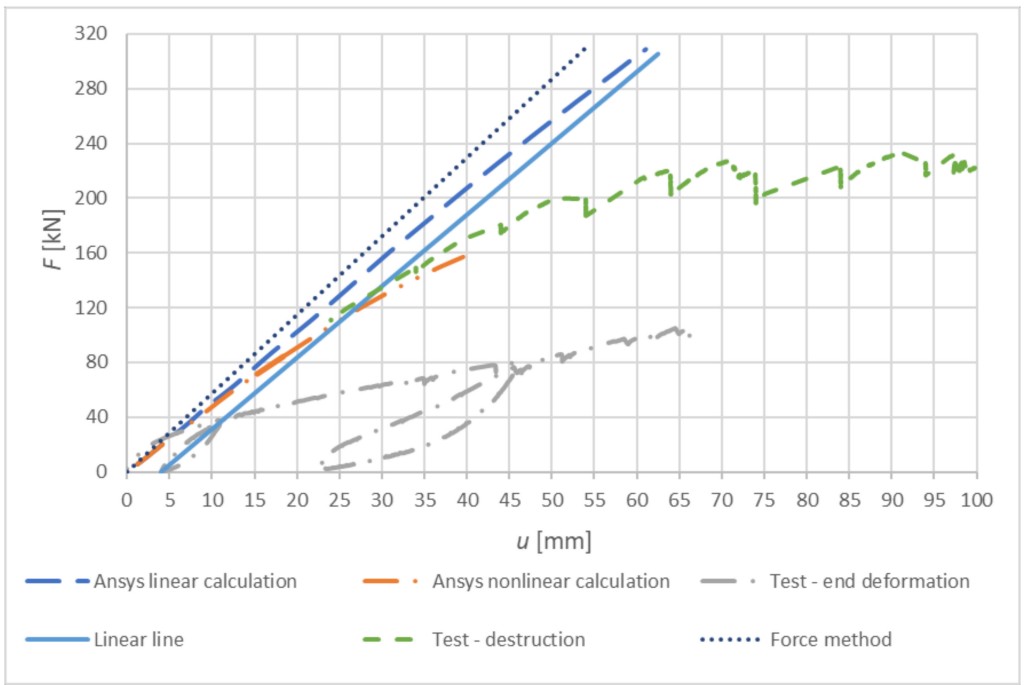

**Figure 15.** Work diagrams of the frame connection.

Testing was also aimed at determining the mode of failure and the force causing the frame connection to collapse. Based on numerical and analytical calculations, it was assumed that the failure in the diaphragm beam was tensile failure perpendicular to the fibers. This hypothesis was confirmed in experimental testing.

The failure of the frame connection occurred in the tension perpendicular to fibers in the top part of the diaphragm beam at a force of $F = 233.30$ kN. In Figure 16, it is possible to see the failure of the diaphragm beam of the frame connection with tension perpendicular to the fibers.

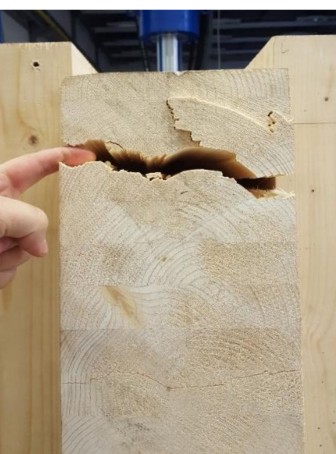

**Figure 16.** Failure of diaphragm beam with tension perpendicular to fibers.

## 4. Discussion

The article was focused on the issue of bending the rigid connection of the diaphragm beam and frame column using mechanical fasteners. In this case, the connection was made using Rothoblaas VGS11400 screws. The research required the creation of numerical models, which were subsequently verified by experimental testing.

The individual research results (manual calculation, numerical models for the Finite Element Model (FEM) and experimental test) were compared with each other using parameters such as the collapse force and the deformation of the timber frame connection.

The values of the maximum load force of the experimental test were compared with the calculation methods shown in Table 1. The value of the calculation according to [28] represented the force at the load capacity for frame connection. The Eurocode 5 [28] value represents 1.56 times less resistance than the load capacity based on experimental testing, which is in good agreement with the results. The highest load value came from a linear numerical model created in ANSYS$^{TM}$, which, however, was not confirmed in experimental testing. On the contrary, the best agreement was represented by a nonlinear numerical model created in ANSYS$^{TM}$, whose value of failure force was closest to the force from experimental testing. This match also indicates a fairly accurate numerical model.

**Table 1.** Comparison of results of individual calculation approaches.

| Method of Calculation | Force Causing Collapse $F$ [kN] | Bending Moment Causing Collapse $M$ [kNm] | Multiplier $M$ |
|---|---|---|---|
| Standard EC5 calculation | 149.10 | 223.65 | - |
| ANSYS-linear calculation | 308.89 | 469.51 | 2.09 |
| ANSYS-nonlinear calculation | 157.94 | 240.07 | 1.07 |
| Experimental test | 230.30 | 349.95 | 1.56 |

Experimental testing has shown that screws that are not commonly used to make this type of joint have sufficient load capacity and rigidity for this use. In addition to the identical rotational stiffness with the standard, a higher load capacity was demonstrated than the load capacity specified in the standard [28]. Such an assumption was expected because the standards calculate, in addition to the maximum load capacity, a certain margin before the collapse of the connection or element. The frame connection of the diaphragm beam and frame column was damaged during testing with tension perpendicular to the fibers at the top part of the diaphragm beam, which supported the accuracy of the numerical and analytical assumptions. The test structure was not reinforced at this point of failure; e.g., by using screws. By reinforcing this, we could expect an increase in load capacity. However, in the order to confirm this hypothesis, it is necessary to perform further experimental measurements with the strengthening of the elements in the tensile element perpendicular to the fibers. In cooperation with practice and research activities, experimental measurements are expected to continue at the Faculty of Civil Engineering of the VSB (Technical University of Ostrava). These experimental tests would aim to compare the effectiveness of bending the rigid connection of the diaphragm beam and frame column formed by screws and bolts, possibly also using bolts and pins.

## 5. Conclusions

Experimental testing is the most concise way to verify the structural details of timber and other constructions. With load tests, it is possible to obtain sufficiently objective and comprehensive knowledge of the action of the connection at a certain nature of load and configuration of the connection in the terms of materials, geometry and design.

The issue of determining the load capacity of the connections of timber structures according to European standards for their design [28] is constantly in development. The

proposed experiment aimed at determining whether rotational stiffness should also contribute to this trend.

The experiment described in this article confirmed the correctness of the numerical models. The non-linear numerical model created in ANSYS$^{\text{TM}}$ software corresponded most to the course of the experimental test. The test also confirmed the correctness of the analytical model, which was calculated using the relationships given in [28–30].

The experiment described in this article proved the suitability of using high-strength screws in the frame connection of a diaphragm beam and frame column. In terms of the load capacity, the experiment demonstrated the safety and reliability of the connection with a certain reserve, as required by [28]. To evaluate the actual rotational stiffness of the connection, it is necessary to perform further tests with the measurement of the deflections of all components of the system.

**Author Contributions:** D.M. and M.J. conceived the experimental measurement techniques; D.M., implementation of numerical models. The experiment was performed under the supervision of M.J., D.M., O.S., L.K., A.L. and P.M., L.K. and M.J. performed the data analysis and the writing of the article; D.M. and A.L. performed the supervision of the experiment and revision of the article. All authors have read and agreed to the published version of the manuscript.

**Funding:** This research was funded by Student Grant Competition VSB—TUO. The project registration number is SP2020/158.

**Acknowledgments:** Experimental measurements and research were carried out thanks to the Department of Structures 221 of VSB Technical University of Ostrava and financial support of companies EXTENT CZ spol. s.r.o., ROTHOBLAAS and INGENIA. Acknowledgments also go to the Department of Building Materials and Diagnostics of Structures 223, Department of Structural Mechanics 228, collaborators of the Centre of Building Experiments VSB—Technical University in Ostrava, the AVC Promotional Centre, Department of Applied Mechanics 330, especially to doc. Ing. Radim Halama, of the VSB—TU Ostrava, Faculty of Mechanical Engineering for the realization of measurements using the IDC image data correlations method.

**Conflicts of Interest:** The authors declare no conflict of interest.

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
