# Peer review of "Analysis of Rotational Stiffness of the Timber Frame Connection"

_sustainability, doi:10.3390/su13010156_

Round 1
Reviewer 1 Report
In my opinion, the paper deserves to be published in the present form.
Author Response
Dear Sir or Madam,
thank you very much for you review. We really appreciate it.
Best regards Marek Johanides.
Reviewer 2 Report
Dear Authors,
Thank you for the prepared material.
In my opinion, the article is written correctly and very interesting.
The article deals with the subject of natural, classical and wooden construction, which is especially popular in American countries (houses in Canadian technology) or in Scandinavia.
An additional aspect is the experimental design and an attempt to analyze the rotational stiffness of the timber frame within the improvement of the frame connection of the timber frame column and the diaphragm beam with using the mechanical fasteners and the issue of determining the load capacity of connections of timber structures according to European standards for the design of timber structures.
The test documentation was correctly made and the correct plots were used (e.g. fig. 13 - as a visualization of the experiment). Of course, concrete remains the most widely used material in the world, but due to the dwindling resources of natural aggregates, other forms of construction are sought or advertised, and wood is an excellent construction material.
Timber frame structures can also be replaced with a steel frame (steel frame), which in turn is conducive to recycling.
I suggest the Authors to compare the wooden skeleton and connections in wooden skeletons, and the steel skeleton in their subsequent analyzes.
It would be advantageous to present the approximate prices of a wooden structure (for comparison).
Below, I am also sending you a few links that suggest what type of comparison I mean.
But these are potential tests for the future within comparisons and sustainable construction and material recovery.
I believe that more attention should be paid to the subject of wooden construction, so I have no objections to the presented work.
Thank You and Best Regards,
Reviewer
https://www.scottsdalesteelframes.com/steel-framing-solutions/panel-fabrication-system?gclid=EAIaIQobChMIg4TG_fnu7AIVReqyCh3QhQmVEAAYASAAEgJ2L_D_BwE
https://www.howickltd.com/applications/commercial-construction/partitions-infill-panels-dry-lining?gclid=EAIaIQobChMIg4TG_fnu7AIVReqyCh3QhQmVEAAYAiAAEgLSx_D_BwE
http://www.stalart.com.pl/
https://sgw-konstrukcje.oferteo.pl/
https://www.sgwkonstrukcje.eu/konstrukcje-stalowe-i-domy-energooszczedne-galeria
Author Response

(The authors gave the same response as above.)

Reviewer 3 Report
This is a re-submitted version of an earlier manuscript (rejected). Authors have made minor changes to the content, making the paper still not suitable for publication, considering the high-quality of the Journal. Authors are referred to the review comments for the earlier version.
comments for original submission:
This paper presents an experimental study on the rotational stiffness of timber frame. Details on the experiment are given, and the results are preliminarily discussed.
Overall, the paper reads more like a technical report rather than a scientific journal paper. The discussions are so limited that the paper offers little insights into the problem. Thus, the reviewer feels sorry but cannot recommend acceptance of this paper.
Author Response
Dear Sir or Madam,
thank you very much for you review. We really appreciate it.
To your comments and recommendations:
The article has been edited according to your previous comments and submitted to the correct edition Civil Engineering as a Tool for Developing a Sustainable Society.
I apologize for the changes, I forgot to inform you earlier
We've added a discussion extension to the article and changed or deleted some images.
We are sorry if the article acts as a technical report, but we tried to formulate them to meet the recommendations of three other reviewers.
The English language has been corrected
Best regards Marek Johanides.
Reviewer 4 Report
This is a re-submission from a previously rejected paper, that I had the opportunity to revise in the original version but also in this new tentative.
It is sad to say that no minimum changes can be noticed from the previous version. The weak points are still there.
Thus I cannot support the publication of the paper in this form.
Comments for original submission:
The paper studies the rotational stiffness of connections in use for timber structures. The topic is generally in line with the journal goals and topics.
However, the current version of manuscript is weak in quality of content and presentation, and in my opinion is not suitable for publication in high-ranked, indexed journal. I will list some aspects only that should be largely improved:
- Starting from some minor aspects, for example, the graphical quality of charts must be improved.
- Often, I can see " Error! Reference source not found " in the main text, and this is another weak aspect that suggests a major revision of the whole document.
- when drawings are presented in figures, the unit of measure must be always defined (in the figure caption or in the figure itself). This is missing for most of the presented drawings
- figure 3 is then redundant: why not presenting 1/4th of the geometry with symmetry planes? It seems a large space that is used for poor information only.
- when the experimental setup and apparatus is presented with figures, actually there are multiple figures for the same subject. These must be merged in sub-items
- the number of figures is also misleading in some cases. Figure 2 appears again after figure 7.
- The scientific content of the paper is then weak due to the lack of appropriate description of methods and appropriate discussion of results.
- for example, I can see very few experimental curves of test results. A comparative study would be beneficial to improve the quality and value of the paper
- a comparative discussion of current results with literature efforts (do they exist for similar structural solutions?) is required to provide an added scientific value
- etc.
Author Response
Dear Sir or Madam,
thank you very much for you review. We really appreciate it.
To your comments and recommendations:
The article has been edited according to your previous comments and submitted to the correct edition Civil Engineering as a Tool for Developing a Sustainable Society.
I apologize for the changes, I forgot to inform you earlier.
-The English language has been corrected
-We've changed the graphic quality of some images
-" Error! Reference source not found " has been fixed
-We added a scale to the drawings
-we deleted figure 3
-We repaired the marking of experimental equipment and instruments
-Figure 2 has been removed
-We have expanded the description of the method
We are sorry if we did not meet all your requirements, but we tried to formulate them to meet the recommendations of the other three residents.
Best regards Marek Johanides.
Best regards Marek Johanides.
Round 2
Reviewer 3 Report
The revision made by authors is incremental. The reviewer feels sorry but cannot recommend acceptance of this paper.
Author Response
Dear Mrs. or Mr.,
Thank you for your review, we appreciate it very much.
I'm sorry if you didn't see all the changes made in the article, there were a lot of them. This was probably due to the fact that the article was completely redesigned in a new template after your first review in the special issue of Sustainable Construction and Building Materials for Environment, and tracking changes was not saved.
In Attachment, I therefore send you the original article sent to Sustainable Construction and Building Materials for Environment for comparison.
In the article were added informations about the numerical model.
We hope that after these edits, you will recommend our article for publication.
Best regards Marek Johanides

Reviewer 4 Report
I'm afraid I cannot still change the final recommendation for this manuscript. Some efforts have been taken, but the weakness still lie in the limited number of experiments, very few data and unproper discussion.
In the "discussion" chapter, "FE modeling" is also cited but I cannot see any numerical model.
Etc.
Author Response
Dear Mrs. or Mr.,
Thank you for your review, we appreciate it very much. In the article were added informations about numerical models. Due to the time and cost of implementation of experimental testing, the article was compared to the data obtained from one experiment and one numerical model. For statistical evaluation, of course, it was necessary to carry out more of these experiments. We are sorry that we do not meet your discussion requirements, but we had to edit the discussion to accommodate the other three reviewers. We hope that after these edits, you will recommend our article for publication.
Best regards Marek Johanides
Round 3
Reviewer 4 Report
.